# Sinuous Antenna for UWB Radar Applications

**DOI:** 10.3390/s22010248

**Published:** 2021-12-30

**Authors:** Luciano Mescia, Gianvito Mevoli, Claudio Maria Lamacchia, Michele Gallo, Pietro Bia, Domenico Gaetano, Antonio Manna

**Affiliations:** 1Department of Electrical and Information Engineering, Politecnico di Bari, Via E. Orabona 4, 70125 Bari, Italy; gianvito.mevoli@poliba.it; 2R&D Department, IAMAtek srl, 70127 Bari, Italy; claudio.lamacchia@iamatek.com (C.M.L.); michele.gallo@iamatek.com (M.G.); 3Design Solution Department, Elettronica SpA, 00131 Rome, Italy; pietro.bia@elt.it (P.B.); domenico.gaetano@elt.it (D.G.); antonio.manna@elt.it (A.M.)

**Keywords:** UWB antenna, miniaturization, passive radar, frequency-independent antennas, supershaped

## Abstract

In this paper, the recent progress on sinuous antennas is detailed, focusing the attention on the antenna geometry, dielectric structure, and miniaturization techniques. In the first part, we introduce the basic principles of the frequency-independent antenna, in particular the self-complementary and log-periodic geometries, as well as the antenna geometries, all characterized in terms of angles. The operating principles, main advantages, system design considerations, limits, and challenges of conventional sinuous antennas are illustrated. Second, we describe some technical solutions aimed to ensure the optimal trade-off between antenna size and radiation behavior. To this aim, some special modification of the antenna geometry based on the meandering as well as on the loading with dielectric structures are presented. Moreover, the cavity backing technique is explained in detail as a method to achieve unidirectional radiation. Third, we present a new class of supershaped sinuous antenna based on a suitable merge of the 2D superformula and the sinuous curve. The effect of the free parameters change on the antenna arm geometry as well as the performance improvement in terms of directivity, beam stability, beam angle, gain, and radiating efficiency are highlighted.

## 1. Introduction

During the last two decades, the technological evolution of mobile communications has strongly changed people’s lifestyles, and the research activities pertaining to the antenna systems design have played a significant role in the development of the so-called wireless revolution. In fact, high-performance antennas for the radio base stations and mobile devices have been the subject of several theoretical studies and practical implementation. Moreover, the growing demand for mobile communication systems has imposed a number of complex and challenging requirements that the UWB antenna system designers have to take into account, such as low cost, low profile, high gain, high efficiency, manufacturer easiness and scalability, integration easiness with well-known microwave circuits, and operation with different types of polarization [1,2].

In addition to the telecommunications, UWB antennas are widely used for a variety of civil, space, biomedical, and military applications [3,4,5,6,7]. They can include UWB pulse radars to detect buried mines or to rescue buried people, as well as the generation of electromagnetic pulse for electromagnetic compatibility and vulnerability tests on electronic and IT equipment [8]. Moreover, during the last few years, they have been implemented in a wide range of military systems requiring spectrum-agility or multifunctionality, such as direction finding systems, electromagnetic jamming, passive radar seekers, antiradiation missiles, and radar electronic support measures [9,10,11,12]. On the other hand, UWB technology is very promising as it is characterized by low vulnerability against the multipath phenomenon, and low probability of detection and interception. In this context, the detection systems based on reflector antennas need UWB antenna feed able to scan a wide frequency range without changing the antenna element. Moreover, in order to improve the receiving capabilities, the antenna should be able to discriminate two orthogonal polarizations.

UWB antennas exhibit a wide impedance bandwidth but not necessarily a stable radiation pattern, as required in most radar applications [13]. Moreover, in addition to the large bandwidth and specific radiation pattern characteristics, the installation space environment usually impacts the antenna design and performance, as it mainly constrains the operating frequency range. As a result, miniaturization techniques need to be implemented to save the installation space and to ensure an optimal trade-off between the antenna size and the radiation characteristics. For UWB radiating elements, this challenge is even harder to fulfill, as gain, beam width, and efficiency should be properly guaranteed. To this aim, several research activities, in both academic and industrial fields, are in progress for the identification of new UWB antenna geometries satisfying the above system requirements and constraints in an increasingly extended frequency band. In particular, they pertain to the planar antennas and they are focused on the improvement of both frequency band and gain uniformity, as well as on the polarization management capabilities, size, and weight reduction. Within this framework, in this paper, we illustrate planar sinuous antennas with conventional and nonconventional arm shapes and operating in the radar frequency band. The paper is structured in a way to highlight the main advantages of these radiating systems as well as the main issues, limits and challenges concerning the operative frequency band, polarization diversity, cross-polarization, gain, input impedance, radiation pattern, and miniaturization. Moreover, it will analyze some technical solutions, special antenna geometries, dielectric structures, and miniaturization techniques aimed to overcome these drawbacks.

## 2. UWB Radar Applications

UWB wireless communication technologies started to receive great attention when the FCC issued a ruling in 2002 that allowed intentional UWB emissions in the frequency range between 3.1 and 10.6GHz, subject to certain restrictions for the emission power spectrum. Such a large bandwidth offered very large data rate, enhanced the signal robustness for data transmission, and allowed the development of radar with high spatial resolution.

In recent years, there has been a relatively large demand for UWB radars for detection, localization, and tracking of living persons behind obstacles for surveillance and rescue operations [14,15,16]. Among different radar sensors, UWB radars have shown some advantages, such as high-resolution ranging in dynamic environments, high performance even under adverse weather and lightning conditions, and low power consumption. On the other hand, electromagnetic waves with ultrawide frequency band can penetrate most common building materials with acceptable attenuation. Therefore, the UWB radars exhibit high performance in multipath channels without requiring LOS. Because of their broadband properties, high-range resolution, and penetrability, the UWB radars can also be used in the field of noninvasive diagnostics and detection of human vital sign signals such as respiration and heart rates [17,18,19]. These systems are used for many other fields, such as nondestructive testing and industrial areas [20], automotive industry [21], impedance spectroscopy [22]. Other interests lie in the areas of Internet of Things and Industry 4.0. Additionally, they are heavily involved in sensor networks, providing high robustness to interference, as well as a low complexity of transmitters and receivers while decreasing the energy consumption.

Taking into account the benefits provided by UWB radars, great effort has been devoted to the research and development in the field of UWB radar devices, antennas, and radar signal processing. Hardware components of UWB radars were presented in [23]. In [24], the authors identified new methods of processing UWB radar signals and possibilities of the UWB radar use in aviation security systems. The authors of [25] present a conceptual guide to the UWB impulse signal and target interaction phenomenon showing how to use the signal spectrum change to identify and enhance detection and tracking of certain target classes. In [26], the authors present a method which uses only one UWB radar device for multiple detection purposes. Moreover, reliability and fault tolerance analysis for the radar device were illustrated, too. UWB radar is widely used for monopulse PRS ARM. It is the main system used in modern electronic warfare to accurately measure the DoA of the electromagnetic wave radiated by the radars and the other radiation sources [5,27]. A problem with DoA estimation is the ability to resolve the closely spaced targets. In [28] is illustrated a method of resolving targets by monopulse radar using Levenberg–Marquardt optimization. The authors of [29] obtained the angular separation of the two closely located targets using diagonal difference channel in the monopulse radar with a four-element antenna system. The study presented in [30] is dedicated to finding a way to reduce errors from estimating the direction of a radar despite the existing distortion features. In this context, the sinuous antennas illustrated in this paper could be good candidates for developing of low-profile antennas capable of transmitting multiple polarizations over ultrawide bandwidths.

## 3. UWB Antennas

UWB radar systems necessitate efficient antennas to provide acceptable bandwidth requirements. Thus, the antenna behavior and performance have to be consistent and predictable throughout the designated UWB spectrum. Ideally, pattern and matching should be stable across the entire bandwidth, and the antenna should preferably have a fixed phase center. Furthermore, the frequency-dependent characteristics of the antennas and the time-domain effects and properties have to be known [31]. There are different types of antennas used in UWB systems, such as planar microstrip, monopole over a metal plate, printed monopole and dipole, stacked patch, tapered slot, metamaterial, dielectric resonantor, TEM horn, and self-similar antennas [1,2,32]. Methods to categorize these antennas can be highlighted in terms of their operating frequency, geometry, function, materials, and so on. Thus, UWB antennas can be roughly classified into two- or three-dimensional, omnidirectional, and directional designs. Several two-dimensional solutions have been proposed in literature referring to different operating conditions in terms of input impedance, polarization, radiation pattern, and bandwidth. A center-fed circular microstrip patch antenna loaded with two annular rings was proposed in [33] to achieve a monopole-like radiation pattern and a wide bandwidth. In [34], the authors illustrated the design of planar differential antenna to be co-integrated with an SoC UWB pulse radar microchip for short-range applications. A miniaturized log-periodic square fractal antenna was presented in [35]. A constant and stable gain in the band 3.1–10.6GHz, as well as a broadside pattern, were achieved. Tapered slot antennas, particularly Vivaldi antennas, are widely used in UWB applications. They are one type of endfire traveling wave antennas. Moreover, a variety of taper profiles and microwave absorbing materials have been presented in literature [36,37]. Other UWB antenna solutions concern wide slot antenna [38], tapered slot antenna [39], and metamaterial antennas [40]. The last class of antennas makes possible the realization of novel functions such as the backward to forward beam scanning including the broadside radiation, the generation of negative-order modes, and the reduction of the surface waves.

Due to their performance in terms of gain, radiation efficiency, and pattern, the LP antennas represent a valid alternative to UWB ones. Moreover, taking into account the low cost, fabrication easiness, and reduced footprint, it is not surprising to guess that they have a huge impact on military and commercial applications. Since their discovery, LP antennas have been the subject of detailed theoretical and experimental studies [41,42]. Several planar LP antennas were proposed in literature. In particular, using arch-shaped cells, it is possible to interleave two or more antennas to radiate with double linear polarization, or combine the two linear polarizations to obtain LHCP or RHCP polarization [43,44]. A wideband cavity backing unidirectional LP antenna is proposed in [45] to obtain enhanced efficiency in a wide frequency range, as well as a better impedance matching at the low-frequency band. Moreover, to simplify the feeding structure and reduce the antenna footprint, a printed LP monopole [46] and trapezoidal dipole [47] antenna array was proposed. Among LP antennas, the sinuous ones have gained lot attention as they can minimize the reflections coming from the antenna ends [48]. Their radiation pattern is similar to that of the spiral antennas, although it can exhibit double polarization and a better uniformity in an extended frequency band. Today, research on sinuous antennas is mostly devoted to the two- [49] and four-arm [50,51] configurations, as well as on slotted ones [52,53,54].

Even if lots of broadband design were proposed, requirements for UWB antennas vary with applications and systems. Horn and reflector antennas are typical three-dimensional and high-gain directional solutions, but their bulky design is not suitable for applications with size constraint, and their directional radiation is not suitable for mobile applications. Conical spiral antennas have frequency-independent impedance response and stable gain over broad bandwidths, but due to the frequency-dependent changes in phase centers, they are not recommended for impulse radio systems. Their nonlinear phase response significantly distorts the waveforms of received pulses by producing undesirable ringing and time delay. From a systems point of view, performance of UWB antennas should be assessed in terms of system transfer function, radiation transfer function, and fidelity, which address the characteristics of transmitting and receiving antennas as well as antenna system response. On the other hand, the development of such systems includes a large number of detailed design topics pertaining to the antenna installation on a platform [55,56]. Firstly, reflections in metal structures on the platform cause a rapidly oscillating ripple in the antenna far-field pattern. Secondary installation effects include diffraction from metal edges and creeping waves on curved metal surfaces. Thirdly, when installing the radome, there is additional ripple due to the combination of radome effects and edge effects. The radome design is typically a trade-off between structural, thermodynamic, aerodynamic, and electromagnetic considerations. As a result, the task for engineers and researchers working on installed UWB antenna performance is to take into account these effects with the aim to find an acceptable antenna placement and installation satisfying the system performance requirements. For example, a modern fighter aircraft may therefore be designed for optimal antenna placement to ensure high electronic warfare capability, with some compromises on the flight performance. On the other hand, some types of UAVs can have small dimensions, so onboard antenna systems should be adequately small. This means that the antenna system should contain a limited number of single antennas located a small distance from each other. Consequently, size reduction became one of the primary objectives of the UWB antenna designers. A number of publications describe sinuous antenna having a large frequency bandwidth in a compact and planar shape, but these solutions do not have extended low-frequency response. Within this framework, in this paper, we illustrate some new designs aimed to analytically modify the sinuous curve in such a way to produce acceptable radiating behavior around 1 GHz and a large frequency band in a very compact installation space environment.

## 4. Frequency-Independent Antennas

When the operating requirements of a UWB antenna require a bandwidth enhancement, gain uniformity, and polarization management flexibility, it is useful to refer to the basic principles characterizing the frequency-independent (FI) antennas. In fact, the geometric properties, such as self-scaling, self-complementary, and self-similarity, ensure that the electromagnetic behavior of these antennas, in terms of input impedance, radiation pattern, polarization, efficiency, etc., is quite frequency-independent or it changes in a predictable and periodic way in a wide frequency range.

It is well known that the electromagnetic properties of the radiating element depend on its dimensions, measured in wavelengths. On the other hand, in the case of a lossless and nondispersive system composed of a mixture of dielectrics and metals, the FI behavior is constrained by the scaling property of Maxwell equations [57]. In other words, if all dimensions of a lossless and nondispersive antenna are scaled by a factor *K*, a particular solution for the fields generated by the antenna remains fixed if the operative wavelength is also scaled by the same factor.

The term FI is, in principle, reserved for radiating elements having no bandwidth limitation. As a result, they have to extend to infinity and, therefore, they do not lead to practical antennas. In order to obtain finite structures matching the physical bound constraints, it is essential to apply some kind of antenna truncation. This operation limits the band over which the performances are almost constant, and in some cases, the performance of the resulting antennas are the same as true FI antennas, but over a limited bandwidth, which can hardly be extended. A way to build practical antennas that are approximately FI over a useful frequency range is to remove any intrinsic scale length by specifying the shape of the antenna only in terms of angles [58]. This method leads, for example, to planar spiral antennas. A second approach is to replicate the antenna cell on discrete scale lengths with tolerable variation in antenna properties for intermediate scale lengths [59]. This scaling behavior can be applied to built log-periodic and related antennas.

### 4.1. Self-Complementary Geometry

The discovery of the principle of self-complementary created a breakthrough in antenna evolution as it provided the theoretical foundation for FI antennas, for antennas with constant input impedance, independent of the source frequency, and it also led to the development of the log-periodic and extremely broadband antennas [60,61,62]. For planar self-complementary antenna (SCA), the conducting and nonconductiong portion of the plane have the same shape and size. Various types of SCA have been developed, and the simplest ones regard two-terminal planar structures where the contour lines of conducting sheets are formed from rotationally symmetric or axially symmetric curved lines with an arbitrary shape [60]. Examples of such SCA are the log-periodic and spiral antennas. However, the principle can be extended to the multiterminal planar SCA excited by a suitable feeding system, as well as to the 3D structures where two infinitely-extended planar conducting sheets comprise a crossing of the vertical and horizontal planes. Examples of such structures are well explained in [60,63]. Even if there is an infinite variety of SCA, the self-complementary properties do not give any useful information about the broadband nature of its radiation behavior and, in particular, it does not ensure radiation pattern independent of the frequency. The SCA, on the other hand, requires an infinitely-extended structure to achieve constant impedance. Thus, the antenna truncation is always needed, and the reduction of the truncation effects is very important in practice. As a consequence, in order to develop practical UWB antenna, it is essential to consider other theoretical issues, such as the Rumsey principle, including information about the radiation characteristics.

### 4.2. Rumsey Principle

A way to define antennas whose pattern and impedance are practically independent of frequency, for frequencies above a specific value, is based on the fact that antenna shapes entirely specified in terms of angles satisfy the scaling principle admitted by Maxwell equations. The general approach is thoroughly explained in [58] and the surface of these antennas can be represented by the general formula
(1)$$r=e^{a(\phi -\phi_{0})}F\left(\theta \right)$$
where r,θ, and ϕ are the standard set of spherical coordinates, F(θ) is an arbitrary function, and *a* and $\phi_{0}$ are two constants. Moreover, as ϕ ranges from −∞ to *∞*, the surface spreads through all space. Equation (Equation 1) is far too general to be practical, and it can be very complicated because an increase of 2π in ϕ does not give, in general, the same *r*. However, practical antenna design having a simple surface which is uniformly expanded in proportion to the distance from the origin can be performed when lnF(θ) is a periodic function of θ with period 2πa. Moreover, if F(θ) satisfies the equation
(2)$$\frac{dF}{d\theta }=r_{0}\delta \left(\theta -\frac{\pi }{2}\right)$$
with δ the Kronecher function and $r_{0}$ an arbitrary constant, then Equation (Equation 1) can be written as
(3)$$\left\{\begin{array}{cc} r=r_{0}e^{a(\phi -\phi_{0})} & \mathrm{if}\;\theta =\frac{\pi }{2}\\ 0 & \mathrm{otherwise} \end{array}\right.$$

Equation (Equation 3) represents an equiangular planar spiral where *a* and $\phi_{0}$ are the expansion rate and orientation, respectively. It is important to outline that the antenna conductors are assumed to be infinitely extended and perfectly conducting, in addition to being surrounded by an infinite homogeneous and isotropic medium. However, antennas of infinite dimension cannot be built, and the inevitable truncation causes the antenna performance to deviate from those obtained for the infinite structure.

### 4.3. Log-Periodic Geometry

The key problem of the antennas all characterized in terms of angles is they have to all extend to infinity because if they do not, they would have at least one characteristic length. As a result, they do not immediately lead to practical designs. However, useful broadband performance can be achieved by relaxing some of the structural requirements, leading to true frequency independence.

The antenna satisfying the Rumsey principle can been seen as a continuum repetition in their geometry bringing a repetition in frequency of their characteristics. Following this principle, an antenna having radiating elements which result from a multiplicity of adjoining cells each scaled in dimension to the adjacent one by a scale factor, is approximately frequency-independent [59]. In particular, for all applicable values of the integer *p*, the following relationship has to be satisfied:(4)$$\frac{R_{p+1}}{R_{p}}=\tau$$
where $R_{p}$ and $R_{p+1}$ represent some dimension of the *p*-th and (p+1)th cells, respectively. The scale factor τ<1 identifies the antenna periodicity in the sense that the antenna proprieties, such as impedance, radiation pattern, etc., are the same every time the frequency $f_{p}$ scales at frequency $f_{p+1}$ by the factor τ or
(5)$$\log f_{p+1}=\log f_{p}+\log\tau$$

Equation (Equation 5) highlights a repetition with a period logτ of the logarithm of the frequency, from which the name log-periodic antennas derives.

The antennas based on the log-periodic geometries operate in a similar way to that of the FI antennas, since their shape may not be entirely defined in terms of angles. Moreover, the features of this type of antenna change moving from a cell and the adjacent one, and the differences become negligible when τ→1. Despite the FI antenna, which is invariably traveling-wave, the log-periodic structure can introduce unintended resonances occurring internal to the antenna arms which degrade performance. On the other hand, log-periodic antennas can be easily truncated on the basis of considerations about the upper and lower frequency cutoff desired for the operational frequency band. The resulting truncated structure is of great practical importance as it yields to in-band performance comparable to the infinite structure when the currents near its ends are sufficiently attenuated.

## 5. Sinuous Antennas

In order to comply with the requests of the UWB radar applications, the radiating elements should be capable of producing (i) UWB radiation in the broadside direction, (ii) dual-polarized radiation, (iii) uniform far-field patterns, (iv) quite constant phase center over the designed band of operation, (v) an essentially frequency-independent input impedance, (vi) symmetrical E- and H-planes radiation pattern, (vii) low SSL and cross-polarization. UWB antenna systems based on bipoles, slots, reflectors, and tapered structures can partially satisfy these requirements, and anyway, they could be heavy, cumbersome, and have low radiation efficiency, as well as they may need complex systems to realize orthogonal-polarized radiation. On the other hand, the spiral antennas could meet most of these requests but they generate RHCP and LHCP polarizations [64]. The dual linearly polarized radiation pattern is an important requirement for polarization diversity radar applications, as well as to enhance the transmitter/receiver isolation. Moreover, the accuracy of short-range radars is improved with the use of polarization diversity [3]. To implement this operation mode, it is important to consider first an antenna arm realizing the linear polarization and then add the same arm pattern, but rotated by 90${}^{{}^{\circ}}$. Moreover, the two antenna patterns should not intersect with each other and, to ensure frequency independent electromagnetic behavior, their geometry should be self-complementary and log-periodic. In this context, the conventional log-periodic antennas are good candidates, but their size can be troublesome. Moreover, the isolation between the two orthogonal polarizations may be deteriorated at higher frequencies, and the copolarization pattern might also became irregular. Owing to these drawbacks, the printed planar sinuous antenna seems to be the most promising option [48]. It can be seen as a combination of spiral and log-periodic antenna concepts which result in a radiating element capable of producing UWB radiation in the broadside direction with polarization diversity. Due to the their unique properties, during the last decades, several types of planar sinuous antennas have been proposed [49,50,52,53,54,65,66]. Non-self-complementary two-arm sinuous antenna and a self-complementary conical projection of the two-arms sinuous structure were introduced to realize unique multiband behavior with alternating polarization handedness between adjacent bands [49]. In particular, the conical structure exhibited unidirectional operation without the use of a cavity backing as well as superior impedance matching and wider bandwidths for all realized bands. A dual-circular polarized dielectric lens-loaded cavity-backed four-arm sinuous antenna was proposed as a transmit antenna in many nontraditional applications requiring higher radiated powers without any external cooling and high efficient high-quality dual-polarized patterns [65]. Sinuous slot antennas were also designed for UWB sensor networks [53] and to allow flush-mountable placement in areas where space is limited [52]. Four-arm sinuous antennas were investigated as a near-field sensing element, such as ground-penetrating radar [51]. Finally, a phased array based on sinuous antennas were presented to improve coupling to telescope optics [67] as well as for L and S band frequencies’ applications [68].

### 5.1. Sinunos Antenna Geometry

The sinuous antenna can be considered as a cascade of *p* cells generated from the sinuous curve. It is defined solely by angles and the growth rate by the equation [48]
(6)$$\phi =\left(-1^{p}\right)\alpha_{p}\sin\left[\frac{\pi }{\ln\tau_{p}}\ln\left(\frac{r}{R_{p}}\right)\right]\;R_{p+1}\leq r\leq R_{p}$$
where p=1,2,…,P and $P\in N_{0}$, *r* and ϕ are the polar coordinates of the curve. $R_{p}$ defines the outer radius of the *p*–th cell, $R_{1}$ and $R_{P}$ being the radius of the outermost and innermost cells, respectively. The *p*–th cell is characterized by the angular width $\alpha_{p}>0$ and the grow rate $0<\tau_{p}<1$. The radii $R_{p}$ are related by
(7)$$R_{p+1}=\tau_{p}R_{p}$$

However, if $\alpha_{p}=\alpha$ and $\tau_{p}=\tau$ are independent of *p*, the curve is a periodic function of the logarithm of the radius *r*. As it is illustrated in Figure 1, one sinuous arm is formed by rotating the curve of Equation (Equation 6) by the angular spacing ±δ around the origin. In this way, the whole sinuous arm oscillates between the two angular limits ±(α+δ). Moreover, the innermost and outermost points of each sharp bend occur at the angle α−δ and α+δ, respectively. Finally, two arcs of angular width 2δ and radius $R_{1}$ and $R_{p}$ are used to outline the innermost and outermost sections of the arm.

The sinuous antennas may consist of *N* arms with a rotational symmetry such that a single arm is duplicated after rotation of 360/N degree about the central axis. In this way, the two arms can be used to create a dipole and to achieve a linear polarization. Moreover, N=4 ensures that the resulting two dipoles rotated by $180{}^{{}^{\circ}}$ do not intersect each other, and that the four-arms sinuous antenna is able to radiate a dual polarized field. For a sinuous antenna, each cell efficiently radiates when the circumferential currents at the beginning and ending of a cell are in phase. As a result, the angular width, angular spacing, and the antenna outer radius $R_{1}$ limit the lowest operating frequency $f_{L}$ which is approximately given by
(8)$$f_{L}=\frac{v}{4R_{1}\left(\alpha +\delta \right)}$$
where the angles are expressed in radians and *v* is the velocity of the wave guided by the antenna structure [69]. However, because of the edge effect due to the abrupt termination of the antenna at the outer resonator, the optimum low-frequency operation is slightly higher than that calculated using Equation (Equation 8). On the other hand, feed point structure sets the high-frequency limit as it is important to provide a good transition from feed point to the active region. In view of this constraint, the highest operating frequency $f_{H}$ is given by
(9)$$f_{H}=\frac{v}{8R_{P}\left(\alpha +\delta \right)}$$

It is important to note that the self-complementary properties are not always satisfied. In fact, for an *N*-arm sinuous antenna, the self-complementary geometry can be obtained only using the specific angular spacing
(10)$$\delta =\frac{\pi }{2N}$$

The circumferential nature of the current distribution arising from the geometrical shape ensures a reasonable *E*- and *H*-plane pattern uniformity. Moreover, due to the interleaved structure, the sinuous antennas have much smaller aperture sizes compared with the conventional log-periodic ones. Another main feature of the *N* arm sinuous antenna is the capability to produce a variety of useful patterns by exciting the antenna in one or more of the normal modes. In particular, all the mode patterns corresponding to the excitation of all *N* arms in the mode *m* have a rotational symmetry and are null on the axis of rotation except for m±1 [48]. Other main features are the low cost, low profile, manufacturing easiness, scalability, and integration with well-known microwave circuits technologies. However, in virtue of Equation (Equation 9), the sinuous antenna could exhibit some technical difficulties in feeding, especially at very high frequencies, due to the very fine details in the innermost part of the structure.

### 5.2. Size Reduction

Any practical antenna should have a finite size to meet certain system size requirements. In fact, the miniaturization process is limited by the fundamental relation between the radiator area and the achievable gain [70]. Once specific values of angular width and spacing are known, by an inspection of Equations (Equation 8)–(Equation 9) it can be noticed that to accommodate the entire band from $f_{\min}$ to $f_{\max}$, the antenna arms have to be truncated, thus setting the values of the radii $R_{1}$ (outermost) and $R_{P}$ (innermost). This truncation is a very important design parameter affecting the antenna size, which can become prohibitively large for most radar applications and unacceptable for small platforms. As a result, during the design of a UWB antenna, the miniaturization techniques need to be considered and explored to improve the low-frequency radiation characteristics, maintaining the same outer dimensions.

The basic concept of the antenna miniaturization involves reducing the wave velocity, modifying the antenna structure in such a way that the local stored electric or magnetic energy density is increased. Within this framework, an inductive loading based on the antenna arm meandering can be used [71]. The procedure of radial modulation was presented in [72]. The resulting self-complementary four-arm trapezoidal antenna structure exhibited a reduction of the lower cut-off frequency without deteriorating the polarization purity. An approach based on meandering has been proposed for improving the low-frequency radiation characteristics of the sinuous antenna [3]. In particular, it was implemented using a cosine function modulating the radial coordinate as
(11)$$\phi =\left(-1^{p}\right)\alpha \sin\left[\frac{\pi }{\ln\tau_{p}}\ln\left(\frac{t}{R_{p}}\right)\right]\;\tau R_{p}\leq t\leq R_{p}$$
(12)$$r=t\left[1+r_{p}\cos\left(\varsigma \phi \right)\right]$$
where ς is the number of meanders in the angular range 0≤ϕ≤2π and $r_{p}$ is relative amplitude of the meandering curve. In this way, the total shape of the conventional sinuous arm can be meandered but the resulting ripples appearing in the inner cells could disturb the high-frequency performance in terms of input impedance and gain. As the cells further away from the central point of the antenna radiate in the lower part of the frequency band, we propose a modified sinuous arm in which just the outermost cell is meandered (see Figure 2) [73]. In this way, the lengths of the current paths within this cell increase, thus improving the low-frequency radiation characteristics.

Material loading is another applicable approach for antenna miniaturization. In this way, the velocity of the current wave can be reduced, thus the antenna appears to be electrically longer. In order to minimize a number of drawbacks concerning practical considerations (cost, size, weight, etc.), a design implementing only dielectric material loading was illustrated in [73]. It is important to point out that some performance degradation is to be expected when proceeding with this step. Consequently, to mitigate these drawbacks, especially at higher frequency, a number of cylindrical rings having electric permittivity $\varepsilon_{i}$ and thickness $d_{i}$, i=1,2,…,N was suitably arranged on the back of the antenna substrate (see Figure 3).

The truncation of the sinuous antenna applied at the end of the last cell generally produces sharp ends. The resulting standing wave around these ends may produce pattern distortion, long ringing tails, and unwanted spikes in the frequency spectrum of the gain [50,51]. To mitigate this problem while improving the gain flatness, a parametric study aiming to identify the optimal truncation of the outermost cell was carried out [73]. Moreover, to perturb the higher operating frequency less, the shape of the arms near the driving points was chosen to be bow-tie pattern. The resulting antenna geometry is illustrated in Figure 4. It has a diameter of 6 cm and exhibits slant-45 polarization as well as wideband performance within the frequency range from 1.5 GHz to 18 GHz. Thanks to the meandering, truncation, and dielectric loading, the minimum working frequency was shifted from 2.9 GHz to 1.5 GHz. Moreover, the optimized boresight gain has a stable behavior inside the interested frequency band, exhibiting a realized gain of about 5 dB.

Figure 5 shows the gain and Figure 6 the squint angle versus the frequency, at the antenna boresight, of the modified four-arm sinuous antenna having the dielectrics loading. In particular, considering ten cylindrical rings, their permittivity and thickness was optimized with the aim to improve the antenna performance at lower frequency. In the figures, the plots pertaining to the horizontal (HP), vertical (VP), and slant-45 (S45) polarization, as well as the typical minimum and maximum limits, are reported. The antenna is fed by an arrangement of orthogonal elements, each feeding a set of opposing sinuous arms. Moreover, to produce S45 polarized radiation, both pairs of opposing sinuous arms were driven by a lumped port set to the impedance of 100 Ω. The whole design was numerically carried out using the 3D electromagnetic simulation software CST Studio Suite^®^. In particular, the frequency domain solver with adaptive meshing option was used. With respect to previous work, we carried out an in-depth analysis aiming to identify some design challenges and to find solutions ensuring better antenna performance. To this aim, a refinement of the antenna shape and footprint was performed. In particular, the chosen dielectric substrate was RT/Duroid^®^ 5880LZ with thickness t=0.127 mm. We selected this material because it has low electrical losses and uniform dielectric permittivity over a wide frequency range. Both τ and $R_{1}$ were adjusted to 0.79 and 30 mm, respectively, in order to fulfill challenging dimensional requirements if compared with the lower working frequency of 1.5 GHz. In addition, taking into account the constraint due to the feeding network, as well as the higher operating frequency ($f_{H}=18\mathrm{GHz}$), the number of cells P, the radius of the bowtie element feed, and the distance between them are kept to constants of 12, 1.65 mm, and 1.2 mm, respectively. It is worthwhile to note that all plots are within the limits set, except some gain values around the lower frequency band.

### 5.3. Cavity Backing

The sinuous aperture is inherently bidirectional and it radiates power in both forward and backward hemispheres about the boresight. In order to convert the bidirectional behavior to unidirectional and to mitigate the electromagnetic interferences due to the installation effects, such as scattering from the mounting surface, the antenna can be backed with a cavity. This solution also allows the flush mounting with mast, fuselage, or hull of the platform [74]. Unfortunately, because of the contamination of the reflected backward radiation, the introduction of the cavity can strongly deteriorate the wideband characteristics of the original sinuous aperture. An empty metallic cylindrical cavity dramatically undermines the antenna performance in terms of radiation pattern, radiation efficiency, and input impedance over multioctave bandwidths [52]. In fact, when the depth of the cavity is $\lambda_{c}/4$, where $\lambda_{c}$ is the wavelength at the frequency corresponding to the midpoint of the bandwidth, the antenna efficiency is two times higher than that of the antenna without reflective cavity. However, for $\lambda \neq \lambda_{c}$, the reflected wave and the electromagnetic wave radiating towards the opposite side of the cavity do not have the same phase, thus the radiation pattern may be badly damaged. On the other hand, a fully loaded cavity with absorbing material attenuates the waves reflected from the bottom of the cavity, damping the cavity resonant modes at the expense of the antenna system efficiency which is reduced by 50%. A dielectric lens-loaded cavity-backed four-arm sinuous antenna was proposed in [65]. The antenna was built without lossy absorber and integrated with a dual-CP beamformer network. Conical configurations have also been presented to achieve unidirectional radiation of waves without the necessity of ground plane or absorptive cavity [75]. The antenna exhibited wide bandwidth and dual circular polarization, but because of the large footprint, it is impractical for the applications where the aerodynamic performance shall not be adversely affected or the installation spaces are limited. Moreover, sinuous antennas on silicon-extended hemispherical silicon dielectric lenses were proposed for millimeter-wave, wideband, dual-polarized radioastronomy receivers [69].

With the aim to prevent contamination of the fields and coupling between the sinuous antenna and cavity backing while minimizing the overall loss, we designed and optimized the cavity configuration illustrated in Figure 7 [76]. The cavity was sized and shaped with the aim to achieve acceptable antenna performance at the lower working frequency of 1.5 GHz. In particular, a multilayer structure having just one layer of absorbing material placed on the bottom of the cavity was considered. In order to obtain good unidirectional performance, the absorber has a nonhomogeneous thickness along the radial coordinate. Moreover, the transmissive layer plays a role in the impedance converter as it better matches the wave impedance of the absorbing layer and free space, reducing the reflected electromagnetic wave by the absorbing material. The whole antenna system has directive stable patterns, good polarization purity, and high realized gain in the frequency band 1.5–18 GHz. Moreover, the installation on the platform is simplified by exploiting miniaturization techniques, allowing a definitive diameter of 6 cm.

Figure 8 and Figure 9 show the gain and the squint angle versus the frequency, respetively, at the antenna boresight. The radiating element is the modified four-arm sinuous antenna with the optimized dielectrics loading. In the figures, the plots pertaining HP, VP, and S45 polarizations, as well as the typical minimum and maximum limits, are reported. In order to facilitate the integration of the feeding circuit, as well as to manage the cavity modes, a truncated metallic cone within the cylindrical cavity was designed. The ARC-PP2000 was used as absorbing material, and the resulting optimized tapered profile has a thickness changing from 6.6 mm to 1 mm. Moreover, the permittivity and thickness of the transmissive layer are equal to 2.8 and 2 mm, respectively. With respect to the free space antenna, the cavity deteriorates the gain around the frequency 6GHz. On the other hand, it eliminates the back radiation and improves the mean gain for frequency higher than 3GHz. Moreover, by an inspection of Figure 9, it can be inferred a stable radiation beam versus the frequency. It is worthwhile to note that the calculated realized gain is higher than that reported in literature and pertaining to an antenna system similar to the one we designed [77]. Moreover, our antenna exhibits a better boresight gain S45 at 2 GHz.

## 6. Supershaped Sinuous Antenna

Many UWB radar application measures, such as direction-finding systems, electromagnetic jamming, passive radar seeker, antiradiation missile, and radar electronic support, often require small radiating element able to provide specific radiation pattern characteristics over a wide bandwidth, spectrum-agility, and multifunctionality properties. In view of these requirements and with the aim to provide a more flexible tool for matching the different needs, we propose innovative sinuous-like shapes, called supershaped sinuous antennas, that make possible the increase of the electrical length under the same footprint [78]. The antenna shape is generated by an analytical formula based on a suitable merge of the 2D superformula [79] and the sinuous curve. In particular, by assuming a Cartesian coordinate system, the sides of the antenna arms are defined by the following parametric equations
(13)$$x=\frac{r\cos[\Phi (r)]}{{\left\{{\left|\frac{1}{a}\cos\left[\frac{m\Phi (r)}{4}\right]\right|}^{n_{1}}+{\left|\frac{1}{b}\sin\left[\frac{m\Phi (r)}{4}\right]\right|}^{n_{2}}\right\}}^{\frac{1}{b_{1}}}}$$
(14)$$y=\frac{r\sin[\Phi (r)]}{{\left\{{\left|\frac{1}{a}\cos\left[\frac{m\Phi (r)}{4}\right]\right|}^{n_{1}}+{\left|\frac{1}{b}\sin\left[\frac{m\Phi (r)}{4}\right]\right|}^{n_{2}}\right\}}^{\frac{1}{b_{1}}}}$$
where
(15)$$\Phi \left(r\right)={\left(-1\right)}^{p}\alpha^{1+\frac{d}{R_{1}}r}\sin\left[\frac{\pi \ln(r/R_{p})}{\ln(\tau )}\right]\pm \delta$$
The parameter *d* allows us to obtain longer or shorter cells, whose end points follow a curve instead of a straight line. Moreover, $a\in R_{0}^{+}$, $n_{q},m_{q},b_{1}\in R^{+}$, q=1,2 are the superformula parameters, where $m_{1},m_{2}$ identify the shape symmetry as well as the number of vertices of the shape. The parameters $a_{1},a_{2}$ control the relative scale of the supershape over each sector. The values of $n_{1},n_{2}$ accentuate the even and odd vertices of the shape, respectively. The parameter $b_{1}$ further determines the shape and it acts as a pull or push force on the sides of the shape. Corners can be sharpened or flattened, and the sides can be straight, convex, or concave. Unlike the conventional sinuous antenna, for the supershaped geometry, the self-complementary property depends on both δ and *m* parameters. The latter identifies the number of sectors in which the plane is folded, so for all *m* values for which the Cartesian plane can be divided in a symmetrical and equal way, the self-complementary condition remains. It is worthwhile to note that the transformation of Equations (Equation 13)–(Equation 15) allows the tailoring of the antenna shape in a simple and analytical way by changing a reduced number of parameters, and it also makes possible the integration in any automated optimization procedure aimed at identifying the shape parameters, yielding the optimal antenna characteristic.

Figure 10 shows a sketch of the supershaped sinuous antenna arm for a specific set of the free parameters. It is formed by rotating the curve of Equations (Equation 13)–(Equation 15) by ±δ around the origin. Moreover, two additional curves defined by the same equations and with angular width 2δ are used to outline the innermost and outermost sections. Some preliminary investigations were reported in [78] with reference to the frequency range from 1 GHz to 10 GHz. The obtained numerical results confirmed that a proper selection of the antenna shape allows the performance improvement in terms of directivity, beam stability, beam angle, gain, and radiating efficiency. Moreover, it was verified that for a diameter of about 6 cm, the supershaped sinuous antenna exhibits better low-frequency radiating properties with respect to the conventional and modified ones.

Figure 11 illustrates the radiation pattern of a two-arms supershaped sinuous antenna in a free space printed on the dielectric laminate RT/Duroid^®^ 5880LZ with thickness t=0.127 mm. Compared with previous work, the antenna geometry was suitably optimized with the aim to improve the radiation properties at the frequency f=1GHz. The resulting geometrical parameters are as follows: m=20; $b_{1}=3$; $n_{1}=4$; $n_{2}=3.3$; d=1; a=b=1. The number of the cells *P*, grow rate τ, and the radius of the outermost cell $R_{1}$ were adjusted to 12, 0.78, and 31.5 mm, respectively. Unlike the modified sinuous antenna, the supershaped ones introduce a significant improvement of the realized gain at 1GHz, exhibiting a value of about 1.1 dB at the antenna boresight.

## 7. Conclusions

In this paper, we illustrated the main advantages, limits, and challenges concerning the planar sinuous antenna-based technologies for UWB radar applications. After describing the operating requirements and principles of these kind of antennas, we analyzed how the geometric properties as self-scaling, self-complementary, and self-similarity can affect their electromagnetic behavior in terms of radiation pattern, input impedance, efficiency, and polarization. Considering the numerous geometric and physical parameters affecting the antenna response, as well as the constraints due to the installation environment, some miniaturization and cavity-backing techniques applied to the sinuous antennas were illustrated. In particular, considering the inductive and dielectric materials loading approach, nonconventional sinuous antennas and special arrangement of high-contrast dielectric materials were presented. Moreover, the lossy cavity-backing technique was outlined with the aim to ensure uncontaminated unidirectional far-field patterns. Finally, the new class of supershaped sinuous antennas was introduced as an innovative sinuous-like shapes that make possible the increase of the electrical length under the same footprint.

## Figures and Tables

**Figure 1 sensors-22-00248-f001:**
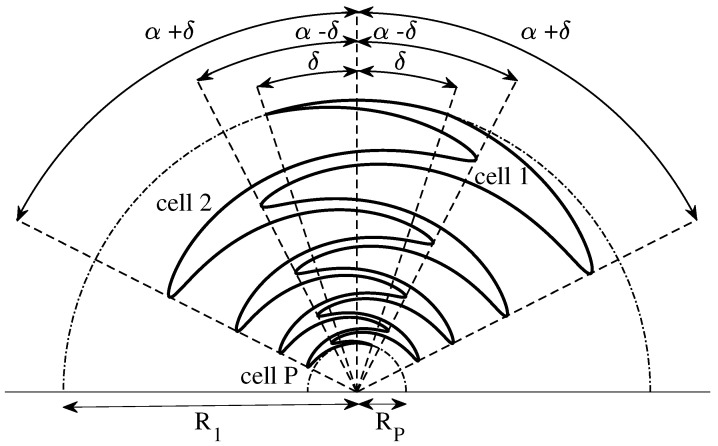
Sketch of the conventional sinuous antenna arm.

**Figure 2 sensors-22-00248-f002:**
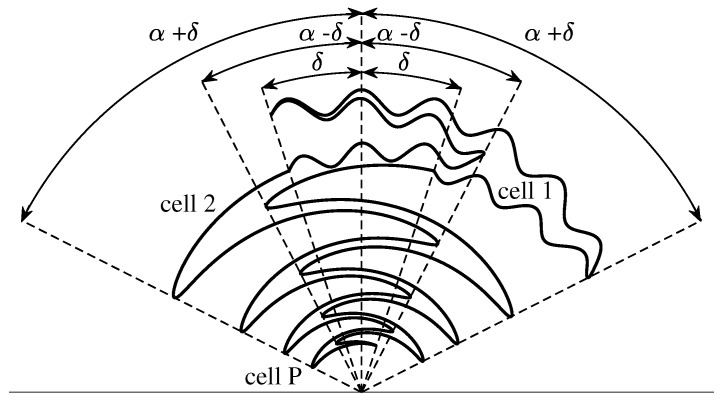
Sketch of the modified sinuous antenna arm.

**Figure 3 sensors-22-00248-f003:**
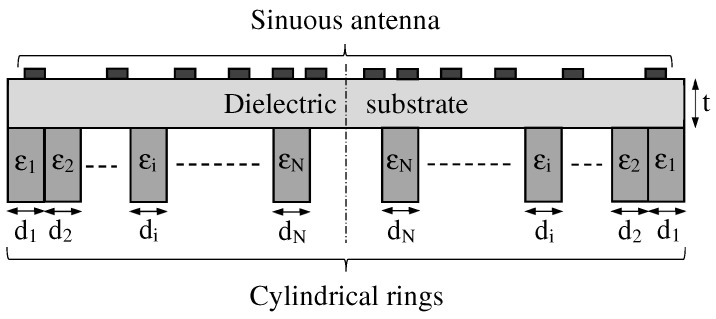
Sketch of the dielectric materials loading the back of the antenna substrate.

**Figure 4 sensors-22-00248-f004:**
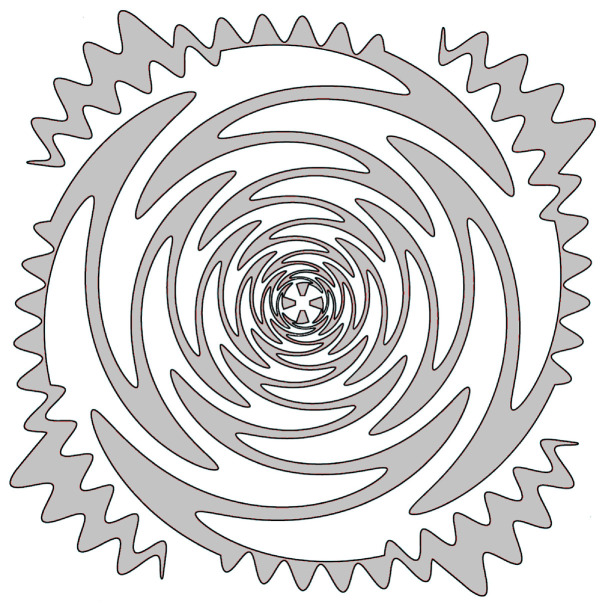
Sketch of a modified four-arm sinuous antenna [73].

**Figure 5 sensors-22-00248-f005:**
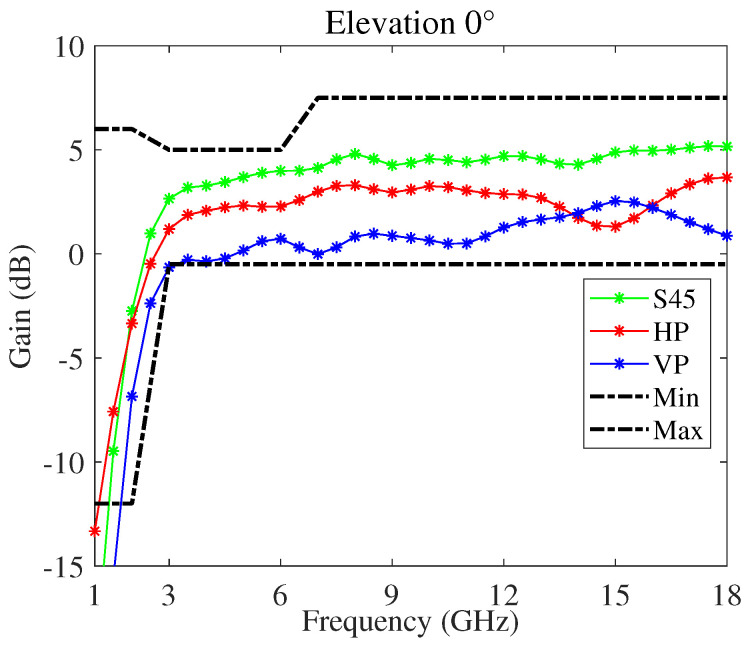
Gain versus frequency, at the antenna boresight, of the modified four-arm sinuous antenna.

**Figure 6 sensors-22-00248-f006:**
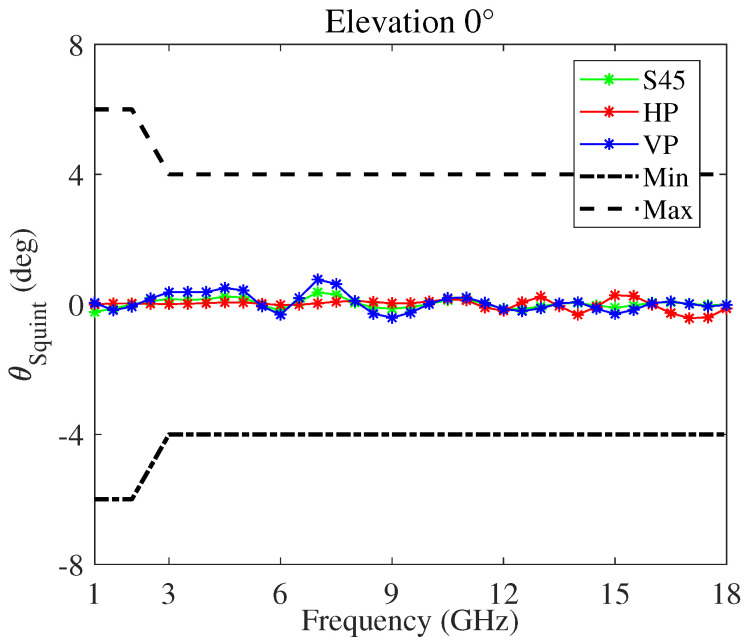
Squint angle versus frequency, with respect to the antenna boresight direction, of the modified four-arm sinuous antenna.

**Figure 7 sensors-22-00248-f007:**
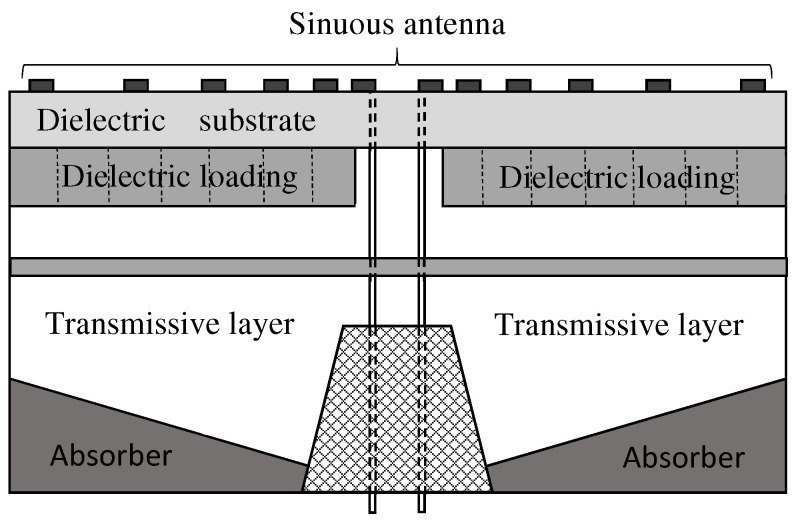
Cross-section sketch of the cavity-backed antenna proposed in [76].

**Figure 8 sensors-22-00248-f008:**
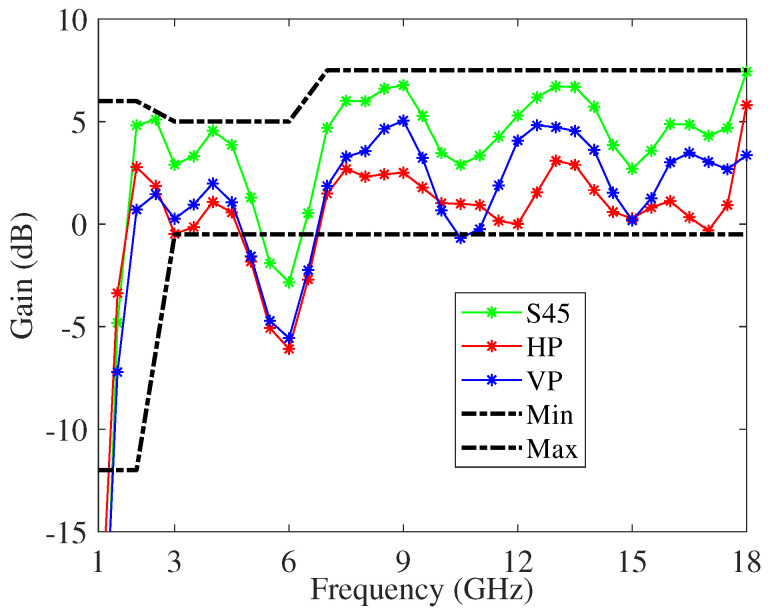
Gain versus frequency, at the antenna boresight, of the modified cavity-backed four-arm sinuous antenna.

**Figure 9 sensors-22-00248-f009:**
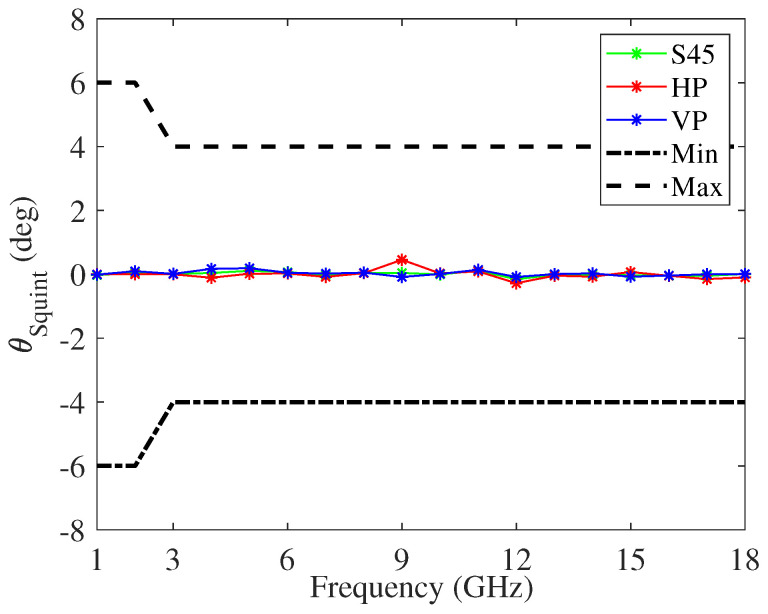
Squint angle versus frequency, at the antenna boresight, of the modified cavity-backed four-arm sinuous antenna.

**Figure 10 sensors-22-00248-f010:**
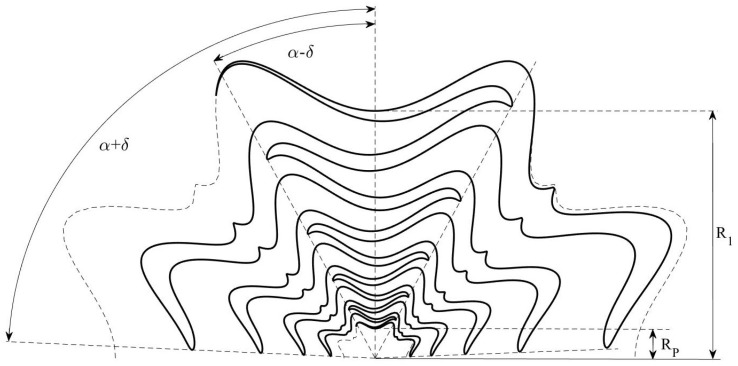
Sketch of the supershaped sinuous antenna arm.

**Figure 11 sensors-22-00248-f011:**
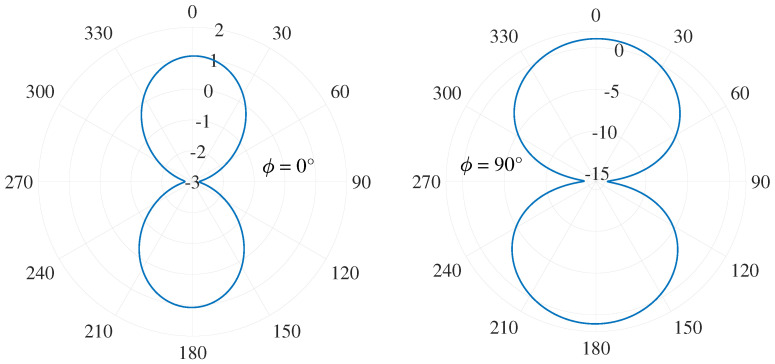
Radiation pattern of printed two-arm sinuous antenna in free space at 1 GHz for both the planes $\phi =0{}^{{}^{\circ}}$ and $\phi =90{}^{{}^{\circ}}$.

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
