# Peer review of "Sinuous Antenna for UWB Radar Applications"

_sensors, 2021, doi:10.3390/s22010248_

Round 1
Reviewer 1 Report
Dear authors,
In this work you provided nice and short overview of the antennas which may be used for UWB devices.
I regard this paper as nice and short overview paper.
I found your work a bit hard to read, although it is well structured.
There are two remarks from my side:
- Provide some simulation results in order better present the concept capabilities.
- Please use shorter and more cohesive sentences.
Best regards,
Reviwer
Author Response
Dear Editor and reviewer,
Thank you for your feedback concerning our paper. You will find below our answers to the reviewer as well as a revised version of our work in which his/her comments have been considered. Below are listed our point-by-point responses to those comments and recommendations.
Reviewer 2
In this work you provided nice and short overview of the antennas which may be used for UWB devices.
I regard this paper as nice and short overview paper.
I found your work a bit hard to read, although it is well structured.
There are two remarks from my side:
- Provide some simulation results in order better present the concept capabilities.
Answer
Thanks for the remark. To this aim, we include in the manuscript some numerical results pertaining to the section 5.2, 5.3, 6. In particular, Figures 5,6,8,9,11 are added and the corresponding explanations are included in the main text.
Please use shorter and more cohesive sentences.
Answer
We revised the manuscript with the aim to give a feedback to your remark
All the change are highlighted using the red color in the revised manuscript
Reviewer 2 Report
The paper presents the theory of a frequency-independent sinuous antenna where the outermost cells are meandered to achieve better directivity, beam stability, beam angle, gain and radiating efficiency. However, no design, simulation, or experimental results of the proposed antenna were presented to validate the claimed achievements of better functionality. I recommend the paper to be rejected and the authors are advised to resubmit with design, simulation, and/or experimental results.
Author Response
Dear Editor and reviewer,
Thank you for your feedback concerning our paper. You will find below our answers to the reviewer as well as a revised version of our work in which his/her comments have been considered. Below are listed our point-by-point responses to those comments and recommendations.
Reviewer 3
The paper presents the theory of a frequency-independent sinuous antenna where the outermost cells are meandered to achieve better directivity, beam stability, beam angle, gain and radiating efficiency. However, no design, simulation, or experimental results of the proposed antenna were presented to validate the claimed achievements of better functionality. I recommend the paper to be rejected and the authors are advised to resubmit with design, simulation, and/or experimental results.
Answer
In order to justify the arguments illustrated in the paper we add Figures 5,6,8,9,11 illustrating some simulation results pertaining modified sinuous antenna in free space, modified cavity backed sinuous antenna and supershaped sinuous antenna.
Reviewer 3 Report
This paper is set to be published as a review of sinuos antenna for uwb radar applications.
The paper needs to be improved to be published as a review of this topic.
- The context of the paper must be highlighted in order to consider the applications in the state of art in uwb radar applications.
- Please add more references. There is a lot of papers dealing with this topic. Please highligh the differences of your paper with respect to previous work cited in the literature.
- A review paper must include a taxonomy or a classification of the previous work. This point is a key aspect of the paper. Please consider it.
- A deep analysis must be done in each study line proposed by the authors, i.e., each uwb antenna must be described detailed with the benefits and advanatages, downsides, study oportunities, etc.
Author Response
Dear Editor and reviewer,
Thank you for your feedback concerning our paper. You will find below our answers to the reviewer as well as a revised version of our work in which his/her comments have been considered. Below are listed our point-by-point responses to those comments and recommendations.
Reviewer 4
This paper is set to be published as a review of sinuous antenna for uwb radar applications.
The paper needs to be improved to be published as a review of this topic.
- The context of the paper must be highlighted in order to consider the applications in the state of art in uwb radar applications.
Answer
You are right. To this aim, a new section “UWB radar application” is added.
- Please add more references. There is a lot of papers dealing with this topic. Please highligh the differences of your paper with respect to previous work cited in the literature.
Answer
Thank you for this remark. In order to give a positive feedback several additional references are added. Moreover, in sections 5.2 and 5.3 and 6 we highlight some difference with respect to previous work.
- A review paper must include a taxonomy or a classification of the previous work. This point is a key aspect of the paper. Please consider it.
Answer
Many thanks for the remarks. To this aim a new section “UWB antennas” was added.
- A deep analysis must be done in each study line proposed by the authors, i.e., each uwb antenna must be described detailed with the benefits and advanatages, downsides, study oportunities, etc.
Answer
In order to give a positive feedback to the reviewer suggestion, we better highlighted some aspects characterizing the sinuous antennas described in the paper.
All the changes are highlighted using the red color in the revised manuscript
Round 2
Reviewer 2 Report
- The title has a typo. It is “Sinuous”, Not “Sinuos”.
- It is not clear if the figure 5-6, 8-9, and 11 were generated using analytical or 3D FEA simulations. If analytical, the authors must mention what equations they have used to generate the figures. If 3D FEA, software name and simulation setup parameters should be mentioned.
- There is no validation or comparison of the results with other published results.
- The authors mentioned” In particular, the chosen dielectric substrate was RT/DuroidR 5880LZ with thickness t = 0.127mm. Both t and R1 were adjusted to 0.79 and 30mm, respectively, in order to accommodate the outer radius according to the reference design.” What is that reference design? Also the rationale for using RT/Duroid 5880 is missing.
- How the cavity specifications were determined?
- How the truncated metallic cone within the cylindrical cavity will be realized in practice?
7. Figure 11 shows radiation pattern at 1 GHz which is beyond the UWB range ( 3.1 and 10.6 GHz).
Author Response
Dear Editor and reviewer,
Thank you for your feedback concerning our paper. You will find below our answers to the reviewer as well as a revised version of our work in which his/her comments have been considered. Below are listed our point-by-point responses to those comments and recommendations.
1. The title has a typo. It is “Sinuous”, Not “Sinuos”.
Reply
The typos was corrected.
2. It is not clear if the figure 5-6, 8-9, and 11 were generated using analytical or 3D FEA simulations. If analytical, the authors must mention what equations they have used to generate the figures. If 3D FEA, software name and simulation setup parameters should be mentioned.
Reply
The simulations were performed using a commercial 3D electromagnetic. In order to clarify this issue, the following sentence was added “The whole design was numerically carried out using the 3D electromagnetic simulation software CST Studio Suite. In particular, the frequency domain solver with adaptative meshing option was used.”
3. There is no validation or comparison of the results with other published results.
Reply
In order to provide a feedback to the reviewer comment the following sentence was added in the revised manuscript “It worthwhile to note that the calculated realized gain is higher than that reported in literature and pertaining to an antenna system similar to the one we designed [77]}. Moreover, our antenna exhibits a better boresight gain S45 at 2 GHz.”
4. The authors mentioned” In particular, the chosen dielectric substrate was RT/DuroidR 5880LZ with thickness t = 0.127mm. Both t and R1 were adjusted to 0.79 and 30mm, respectively, in order to accommodate the outer radius according to the reference design.” What is that reference design? Also the rationale for using RT/Duroid 5880 is missing.
Reply
In order to avoid misunderstanding, the following sentences are included in the revised paper “in order to fulfill challenging dimensional requirements if compared with the lower working frequency of 1.5 GHz” and “We selected this material because it has low electrical losses and uniform dielectric permittivity over wide frequency range”
5. How the cavity specifications were determined?
Reply
To provide a feedback to the reviewer comment the following sentence was included in the revised manuscript “The cavity was sized and shaped with the aim to achieve acceptable antenna performance at the lower working frequency 1.5 GHz”
6. How the truncated metallic cone within the cylindrical cavity will be realized in practice?
Reply
The truncated cone may be realized by using a sheet metal stamping technique:
7. Figure 11 shows radiation pattern at 1 GHz which is beyond the UWB range ( 3.1 and 10.6 GHz).
Reply
Typical technical characteristic required for Radar-Electronic Support Measurements equipment is a wide electromagnetic spectrum (total bandwidth) capability (typically 0.5 to 40 GHz). In view of this requirement we have focused our attention on the antenna design working at frequencies beyond the typical UWB range. However, we select the radiation pattern at 1 GHz because at such frequency the antenna design challenging especially by considering the reduced antenna footprint.
All the change are highlighted using the blue color in the revised manuscript.
Reviewer 3 Report
The authors have addressed the reviewers' comments. The paper can be published.
Author Response
Many thanks for the positive feedback
Round 3
Reviewer 2 Report
Minor English editing is necessary.